# Validation of an Automated Body Condition Scoring System Using 3D Imaging

**Niall O' Leary** [1,*], **Lorenzo Leso** [2], **Frank Buckley** [3], **Jonathon Kenneally** [3],
**Diarmuid McSweeney** [4] **and Laurence Shalloo** [3]

1   Hincks Centre For Entrepreneurial Excellence, Cork Institute of Technology, Cork, T12 P928 Co. Cork, Ireland
2   Department of Agricultural, Food and Forestry Systems, University of Florence, 50145 Firenze, Italy;
    Lorenzo.Leso@unifi.it
3   Moorepark Animal & Grassland Research and Innovation Centre, Teagasc, Fermoy,
    P61 C997 Co. Cork, Ireland; Frank.Buckley@teagasc.ie (F.B.); Jonathon.Kenneally@teagasc.ie (J.K.);
    Laurence.Shalloo@teagasc.ie (L.S.)
4   Unit 2, True North Technologies, Shannon Business Centre, Shannon, V14 YT99 Co. Clare, Ireland;
    Diarmuid.McSweeney@moregrass.ie
*   Correspondence: Niall.OLeary@cit.ie or NiallOLeary@gmail.com

**Abstract:** Body condition scores (BCS) measure a cow's fat reserves and is important for management and research. Manual BCS assessment is subjective, time-consuming, and requires trained personnel. The BodyMat F (BMF, Ingenera SA, Cureglia, Switzerland) is an automated body condition scoring system using a 3D sensor to estimate BCS. This study assesses the BMF. One hundred and three Holstein Friesian cows were assessed by the BMF and two assessors throughout a lactation. The BMF output is in the 0–5 scale commonly used in France. We develop and report the first equation to convert these scores to the 1–5 scale used by the assessors in Ireland in this study ((0–5 scale × 0.38) + 1.67 → 1–5 scale). Inter-assessor agreement as measured by Lin's concordance of correlation was 0.67. BMF agreement with the mean of the two assessors was the same as between assessors (0.67). However, agreement was lower for extreme values, particularly in over-conditioned cows where the BMF underestimated BCS relative to the mean of the two human observers. The BMF outperformed human assessors in terms of reproducibility and thus is likely to be especially useful in research contexts. This is the second independent validation of a commercially marketed body condition scoring system as far as the authors are aware. Comparing the results here with the published evaluation of the other system, we conclude that the BMF performed as well or better.

**Keywords:** Body condition score; cows; automated; validation; precision technology

## 1. Introduction

Formalised body condition scoring (BCS) scales classify cows on a range from emaciated to obese. Cow health and performance is strongly associated with BCS. For example, infertility, metabolic disorders, and lameness rates increase when cow condition deviates from the recommended BCS range [1,2]. Manual BCS scorers visually assess cow body shape and/or palpitate defined anatomical regions—the specifics vary by scale [3]. However, assessor skill and subjectivity limit sensitivity to smaller differences in condition and reproducibility even among highly trained scorers [2,4,5]. In this context, reproducibility relates to the extent that one scorer will be consistent with themselves when assessing the same animal at a later time. In contrast, how consistent a method is relative to another method at approximately the same time is called agreement. In addition to reproducibility of data collected, data collection itself is limited by the time intensive nature of manual scoring. An automated

system once installed could score cows regularly, increasing the quantity of data [2,6]. However, the automated systems' reproducibility should also be similar or greater than manual observers. If the system can collect data regularly and with high reproducibility, it will be of value to farmers and researchers. One potential advantage is that cow management could be individualised. If cow BCS gradually extends beyond the recommended range, management (e.g., feed supplement levels and milking frequency) could be adjusted with the aim of returning the cow to a healthy BCS. Rapid changes in BCS could generate health alerts, thus the farmer and the vet could examine the cow.

Analysing 3D images of cows' backs taken from above is a common approach in the literature to measure cow condition [7–9], and recent research has reported accuracy improvements using machine learning techniques [10]. To date, only three automated BCS systems have been marketed commercially. All three systems use image analysis-based approaches (DeLaval (Tumba, Sweden), Ingenera SA (Cureglia, Switzerland) and Biondi Engineering SA (Cadempino, Switzerland)). Only one validation study has been published examining the "DeLaval Body Condition Scoring, BCS DeLaval International AB, Tumba, Sweden" [11]. The system was found to be accurate for moderate BCS scores but was in poor agreement with human observers for both relatively low and relatively high values [11]. The magnitude of deviation from normal ranges was thus unreliable. This indicates the system would be useful for automated management to increase or decrease BCS back to a desired range but may not be useful for identifying extreme cases that might need prompt veterinary examination. In this study, another commercially available system, the BodyMat F (BMF, Ingenera SA, Cureglia, Switzerland), is assessed for reproducibility and agreement with two expert BCS scorers.

## 2. Materials and Methods

### 2.1. Data Collection

As the study was observational in nature or using data already routinely collected with no invasive procedures, formal ethical approval was not required. Data collection occurred at Dairygold research farm, Fermoy, Cork, Ireland in 2016. The BMF was fitted to a drafting gate system at the exit of the milking parlour. Image acquisition was triggered by reading the cows' electronic ID tags. Using either a 2D image with a laser line projected on the back (this study) or a 3D image (later versions), back shape was measured. The BMF uses back shape to estimate BCS on a 0–5 BCS scale commonly used in France in units of 0.1 [12]. One hundred and three spring calving, mostly Holstein Frisian cows, were regularly assessed by the BMF, resulting in 1945 records. Two highly trained BCS scorers, Frank Buckley (FB) and Jonathon Kenneally (JK), over 10 and 6 sessions assessed cows, creating 560 and 476 BCS score records, respectively, on a 1–5 scale commonly used in Ireland [3,13]. To ensure that scores were assigned independently, the assessors did not discuss scoring of individual cows. Manual assessments for each cow were recorded using a handheld device, which also read the Radio Frequency Identification tag on each cow. The resulting report contained the cow identification, the timestamp, and the BCS as assessed by each assessor.

### 2.2. Data Analysis

Statistical analysis was carried out using R [14]. A data frame was built combining the manual assessment and the BMF scores by week. We publish this data here [15]. Averaging the score of assessors reduces variability [7,16], and the average/mean of two assessors constitutes the benchmark for the BMF in this study. The BMF output is in the French 0–5 BCS scale [12]. However, the assessors used a 1–5 scale commonly used in Ireland [3]. To allow comparisons, we converted the BMF output to the 1–5 scale by using a simple linear model and by using the mean of the two observers as the dependent variable. This is not ideal, as it means that linear over or underestimation of BCS by the BMF is not discernible from this analysis, as it would be obscured by the conversion process. However, a linear bias would be relatively simple to adjust for and would not undermine the usefulness of the

tool assuming the tool could still differentiate higher condition cows from lower condition cows (rank) and can quantify differences.

To assess agreement between the converted BMF output and the mean of the two human assessors, three methods are used. Firstly, Bland Altman plots are used with an adjustment applied to account for repeated measures [17,18]. Secondly, Lin's Concordance of Correlation Coefficient (CCC) is used, which assesses agreement between assessors Frank Buckley (FB) and Jonathon Kenneally (JK), and between the mean of FB and JK, and the BMF. In this study, as cows were scored multiple times, a CCC method adapted for repeated measures was implemented using the "CCCrm" R package [19]. Thirdly, to allow comparison with another similar study [11], Pearson's R is also reported. We also assess reproducibility of the methods within subject or the extent the methods provide the same scores at different times for the same subject. This is done by calculating the standard deviation a month apart along with the 95% reproducibility limit as per the calculations described in McAlinden et al [20]. The data and analysis scripts are published here [15].

## 3. Results

Table 1 reports the number of cows assessed by each method during each week with manual observer records. In total, FB recorded 560 scores and JK recorded 476. There were 426 matches where FB and JK scored the same cow in the same week. The BMF had 1945 scores (max—one score per cow per week).

**Table 1.** Count of records from manual observers FB and JK, and the BodyMat F (BMF).

|  | Apr 12th | Apr 26th | May 10th | Jun 7th | Jul 19th | Aug 16th | Oct 11th | Nov 1st | Nov 8th | Dec 6th | Total |
|---|---|---|---|---|---|---|---|---|---|---|---|
| FB | 97 | 5 | 93 | 92 | 66 | 80 | 62 | 62 | 0 | 3 | 560 |
| JK | 97 | 0 | 92 | 91 | 70 | 80 | 0 | 0 | 46 | 0 | 476 |
| BMF | 102 | 5 | 99 | 94 | 72 | 82 | 65 | 62 | 47 | 3 | 1945 * |

BMF: BodyMat F. * Total records relate to all of 2016 including weeks when no matching manual observation was recorded.

Cows' average parity was 2.54. Their average BCS score was 2.89 as measured by the manual observers as of the week of 12 April 2016 (Table 2). This was the first week of the trial and the week with the most records. Numbers reduced subsequently due to tag loss or animals leaving the herd. A linear model (lm function) was calculated in R with the Body Matt F output on the french 0–5 scale as the independent variable and the mean of the two human assessors (1–5 scale used in Ireland) as the dependant variable. The following equation was found to best convert between scales: 0–5 French scale $\times$ 0.38 + 1.67 $\rightarrow$ 1–5 Irish scale and the adjusted coefficient of determination ($R^2$) for this linear model was 0.52.

**Table 2.** Descriptive statistics as of 12 April 2016.

|  | Mean | Median | Min | Max | 1st Qu. | 3rd Qu. |
|---|---|---|---|---|---|---|
| Days in milk | 62.39 | 62 | 36 | 90 | 55 | 71 |
| Cow lactation | 2.6 | 3 | 1 | 5 | 1 | 4 |
| FB assessor BCS scores | 2.9 | 3 | 2.5 | 3.5 | 2.75 | 3 |
| JK assessor BCS scores | 2.94 | 3 | 2.5 | 3.5 | 2.75 | 3 |
| Mean assessor BCS scores | 2.89 | 2.91 | 2.44 | 3.24 | 2.8 | 2.98 |
| BMF BCS scores (converted) | 2.92 | 3 | 2.5 | 3.5 | 2.75 | 3 |

BCS: body condition score. Qu: Quartile.

Table 3 reports that, within subjects, standard deviation was lower for the BMF relative to the mean of the manual observers and much lower than compared to individual observers. The BMF was more likely to score the same cow as having the same BCS a month later than the human observers.

Table 4 presents the CCC analysis between data sources. The agreement between the BMF and the assessor mean was 0.67, equivalent to the agreement between assessors.

Figure 1 presents the frequency distributions and the Bland–Altman plots for the mean of assessors and the BMF. Comparing Figure 1A,B and examining Figure 1D demonstrates that, relative to the human mean scores, extreme values were attenuated by the BMF, particularly for cows of condition greater than 3.25 as measured by the mean of the two human observers. The agreement between the two human observers showed no nonlinear bias (Figure 1C). In contrast, Figure 1D shows that the BMF relative to the mean assessor score attenuated higher BCS scores.

**Figure 1.** Frequency distributions of BCS scores for the mean of both manual assessors (**A**) and the BMF (**B**). Bland–Altman plots for manual assessors' agreement with a lower confidence limit of −43, a mean difference of −0.11, and an upper confidence limit of 0.2 (**C**) and the mean of the assessor's agreement with BMF with a lower confidence limit of −0.3, a mean difference of 0 (due to the scale conversion), and an upper confidence limit of 0.3 (**D**).

**Table 3.** Within subject standard deviation (95% reproducibility limit) as per calculations described in McAlinden et al [20] (*n* = 83).

|  | **Week of 12 April to Week of 10 May** | **Week of 10 May to Week of 7 June** |
|---|---|---|
| BMF | 0.006 (0.016) | 0.004 (0.012) |
| Mean of FB and JK | 0.009 (0.026) | 0.007 (0.022) |
| FB | 0.012 (0.033) | 0.012 (0.033) |
| JK | 0.013 (0.037) | 0.010 (0.026) |

**Table 4.** Concordance of correlation (CCC) (95% confidence interval) having accounted for repeated measures and the Pearson's correlation (r) for the week of 12 April 2016 (no repeated measures) for comparison with Mullins et al. [11].

|  | **FB** | **JK** | **JK FB mean** | **FB** | **JK** | **JK FB Mean** |
|---|---|---|---|---|---|---|
|  | CCC | CCC | CCC | r | r | r |
| JK | 0.67 (0.59–0.73) |  |  | 0.76 |  |  |
| BMF | 0.57 (0.47–0.66) | 0.57 (0.47–0.66) | 0.67 (0.58–0.75) | 0.67 | 0.69 | 0.72 |

CCC: Concordance of correlation, r: Pearson's correlation.

## 4. Discussion

This study is the first to report an equation for converting the 0–5 scale used in France to the 1–5 scale used in Ireland as far as the authors are aware. The $R^2$ of 0.52 for the conversion model was at the lower end of the range of $R^2$ values Roche et al. [3] reported when comparing four BCS scales. This indicates it will be of use for automated BCS systems converting their output between these two scales and comparing scientific literature using these scales. However, it is likely the conversion could be improved by following a methodology similar to Roche et al. [3]. Because of this conversion, we are not able to comment on if the BMF over or underestimates BCS in a linear manner, as this was obscured by the need to convert scale. However, a linear bias would be relatively simple to adjust for. We can, however, discuss the ability of the BMF to rank cows from low to high condition, its ability to detect extreme values, and potential nonlinear patterns of disagreement or bias in the scoring.

The inter-assessor agreement was 0.67 (CCC), and the BMF-assessor mean agreement was also 0.67. Both indicate only moderate agreement [21]. This level of agreement between observers is a limitation of this study, as it indicates our gold standard/reference internal agreement could have been greater. The reported agreement with the BMF thus needs to be interpreted relative to the agreement within the gold standard/reference. Ideally, our observers could have scored on the same day, but this was logistically burdensome and would have curtailed the data collection. The method employed was a trade-off resulting in a relatively large amount of repeated measures throughout a lactation, allowing us to assess method reproducibility. Furthermore, the assessors could have calibrated their scoring. The two scorers did not calibrate their scoring to improve their agreement for this trial. They had last done so 16 years previously in 2000 [22], likely explaining the moderate agreement. In this regard, both scorers were as independent to each other as two scorers might be from two independent research institutions.

These issues highlight how reproducibility issues might arise between institutions and the potential value of an automated method with high reproducibility. Future work should prioritize human observer calibration or could look at carcass fat composition as a better gold standard indicator of condition. This work could dovetail with technology assisted selection of cattle on farm for slaughter [23]. The BMF was more likely to score the same cow as having the same BCS a month later than the mean of the assessors and the assessors individually as measured by within subject standard deviation (Table 3). This indicates the BMF output was more reproducible. In this context, reproducibility relates to the extent that one scorer will be consistent with themselves when assessing the same animal at a later time. Another potential explanation is that the BMF was less sensitive to real changes in condition within cows than the mean of the assessors. Cow condition does tend to change over a lactation [2]. The BMF-assessor mean agreement and assessor-assessor agreement as measured by CCC were the same (0.67). Therefore, if the BMF was less sensitive to changes within a cow, it would imply a correspondingly higher sensitivity to differences between cows to end up with the same level of agreement as the human assessors, which seems unlikely. Overall, the most convincing interpretation was that the BMF had greater reproducibility than assessors individually and the mean score of both assessors. The BMF will thus likely be of particular interest to researchers where reproducibility of findings and regular BCS assessment is a priority.

Mullins et al. [11] assessed the commercially available DeLaval Body Condition Scoring system. They used a differing scale [24] than used here [13], thus some caution when making comparisons is required, but both are 1–5 scales with increments of 0.25. Mullins et al. [11] presented a Bland–Altman plot with 95% confidence intervals (two standard deviations) of agreement of approximately 0.5 to −0.7. The equivalent interval in this study was ±0.3. This indicates the BMF was more in agreement with manual observers than the DeLaval system by this measure. In fact, the standard deviation in BCS score disagreement was half that reported for the DeLaval system.

In comparison to Mullins et al. [11], the Pearson's r between the gold standard and the BMF was lower (0.72 vs. 0.78). However, in light of the lower within gold standard correlation here (0.67 vs. ~0.86), the BMF camera to mean of assessor correlation looks more positive. Mullins et al. [11] reported that the DeLaval system was less accurate for very under- and over-conditioned cows attenuating scores towards the mean [11]. A similar but weaker attenuation effect was observed for the BMF, but only for over-conditioned cows. This indicates that the BMF is more sensitive to cows with lower condition scores (Figure 1). Both systems thus appear to struggle to accurately score over-conditioned cows. This may be due to reduced variation in back shape with increasing fat deposition. Determining fat deposition is why some BCS scales require palpitation of the back around the tail bone, as it is not easily discerned visually [2]. This would indicate that 3D imaging technologies are likely to be in greater agreement with BCS scales that do not require palpitation.

The BMF would appear to be superior at identifying cows that are very low condition and thus may need to be examined by the farmer or the vet. This is because, while the DeLaval system could discern a deviation from the recommended BCS range, Mullin's et al. [11] indicate the magnitude of the deviation, and thus potential urgency of assessing the cow could not be reliably discerned. Both the present study and Mullins et al. [11] illustrate that there are now commercially available products that can assess BCS automatically. The present study indicates that, on balance, comparing these two evaluation studies, the BMF performed as well if not better than the DeLaval system. Both studies were carried out exclusively with Holstein Friesian cows. Therefore, further validation with other breeds is required [2].

## 5. Conclusions

In summary, the BMF agreement with the mean assessor scores was equivalent to inter-assessor agreement and was more consistent. This indicates the BMF is as robust a BCS measure as either assessor alone, illustrating the potential value of an automated and non-subjective BCS assessment method. This is the second independent validation of a commercially available automated body condition scoring system. Relative to the other system, the BMF output appears to regress less to the mean, being more sensitive to variation in BCS, in particular for lower condition score cows. In conclusion, the automated, non-subjective nature of the BMF combined with the ease of collecting regular scores indicate the BMF would likely be of value in commercial and research contexts assessing Holstein Friesian cows.

**Author Contributions:** Conceptualization, L.L., and L.S.; methodology, N.O.L., L.S.; software, N.O.L.; validation, N.O.L., L.S.; formal analysis, N.O.L.; investigation, L.L., D.M., F.B., and J.K.; resources, L.S.; curation, N.O.L.; writing—original draft preparation, N.O.L.; writing—review and editing, N.O.L., L.L. and L.S.; visualization, N.O.L.; supervision, N.O.L. and L.S.; project administration, L.S.; funding acquisition, L.S. All authors have read and agreed to the published version of the manuscript.

**Funding:** This research was funded by Science Foundation Ireland, grant 13/IA/1977 (PrecisionDairy) as well as Science Foundation Ireland and the Department of Agriculture, Food and Marine on behalf of the Government of Ireland under the Grant 16/RC/3835 (VistaMilk).

**Acknowledgments:** The assistance of Jessica Werner (formerly Teagasc and now Hohenheim University, DE) and Andrea Biondi (Biondi Engineering SA, CH) in this study are gratefully acknowledged.

**Conflicts of Interest:** Fifth author, Diarmuid McSweeney, was a Teagasc technician who in 2016 contributed to the collection of the data used in this study. He was subsequently employed by Ingenera SA between February 2017 and February 2018.

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
