# Peer review of "Validation of an Automated Body Condition Scoring System Using 3D Imaging"

_agriculture, doi:10.3390/agriculture10060246_

Round 1

Reviewer 1 Report

Review

The study is interesting and important because it applies authomatic systems for evaluation of BCS and in future it opens new posibilities in managing dairy cows (scanning of BCS every milking – modelling through the whole lactations, etc.)

This manuscript was second study of this type (according to the authors), so we are at the beginning. Manuscript open many new questions and problems and it is necessary to solve  in next papers (hope will be).

Also interesting could be use scanning for estimated exterieur traits or body weight.

From my point of view problematic is poor correlation between assessors (CCC 0.67). For example, we have regular „calibrating meeting“ of linear type traits evaluators every year in Czech Republic. Also it is necessary compare separately for different breeds (as you wrote not only Holstein on farm).

On the other hand I don´t see a problem with converting scale using simple linear model. Adj. R2 only determine how many percent of variation we explain by the model, sometimes poor R2 not examine poor model. In fututre research it is necessary to analyse possibility of incorrect rating of under or overconditioned cows. It coul be interesting to combine results with change of body weight (assumed weighing-machine in milking parlour pasageways).

Question for authors

  1. Which breed (except Holstein) were on farm?
  2. Authors describe problem with palpitation which is not possible in scanning by camera by overconditioned cows. Maybe better to change methodology of BCS evaluation in fututre? What is author opinion?
  3. CCC, r and SD are computed only for animals on one specific lactation or through all animals in all lacation on farm?

Another comments

Last paragraph (lines 223 – 230) should be separate chapter called „Conclusion“.

Please check spelling and other technical issues (for example in sentence below table 1, which begins *, there is 2 times word „include“).

Finally the manuscript is written in a clearway, also the article is actuall and leads to very important breedin gissues. I propose to publish this manuscript.

Author Response

Thank you for the constructive comments.

Which breed (except Holstein) were on farm?

You are right to raise the issue of breed. The authors have tried to test the validity of another BCS system where there was a cross-bred animals and achieved very poor results (and so did not continue to publication due to lack of cooperation from the system vendor).  We have added the following sentence to the discussion.  ‘Both studies we carried out exclusively with Holstein Friesian cows and further validation with other breeds is required [2].’ Ln 228-231.

Authors describe problem with palpitation which is not possible in scanning by camera by overconditioned cows. Maybe better to change methodology of BCS evaluation in fututre? What is author opinion?

There is no clear answer to what is most appropriate. Palpitation is arguably important for manual observation but some BCS scales do not include it. The suggestion/observation that a camera system will likely perform better when compared to a manual scale without palpitation, and so this should inform the interpretation of results using different manual scales, is the extent of our insight currently.

CCC, r and SD are computed only for animals on one specific lactation or through all animals in all lacation on farm?

All animals calved in spring (January – March). Data collection began in April and continued through to November December so most of one lactation was included.

Last paragraph (lines 223 – 230) should be separate chapter called „Conclusion“.

Implemented.

Please check spelling and other technical issues (for example in sentence below table 1, which begins *, there is 2 times word „include“).

Implemented

Reviewer 2 Report

Leary et al report the validation of an automated Body Condition Scoring system using 3D imaging. An interesting work, for evaluating a commercial device. Authors may pay attention to some minor points that can offer added value to the manuscript.

line 84 proprietary algorithm. What does this mean? authors may specify this algorithm even with a supplementary file.

Authors should make clear the practical benefit of such an evaluation and also any future perspective that present work may assist.

Author Response

Leary et al report the validation of an automated Body Condition Scoring system using 3D imaging. An interesting work, for evaluating a commercial device. Authors may pay attention to some minor points that can offer added value to the manuscript.

Thank you for the constructive comments.

line 84 proprietary algorithm. What does this mean? authors may specify this algorithm even with a supplementary file.

The algorithm is property of the vendor and the authors do not have access to it. We have rewritten the sentence to make it clearer. ‘ The BMF uses back shape to estimate BCS on a 0-5 BCS scale commonly used in France in units of 0.1 [12].’ LN87 & 88.

Authors should make clear the practical benefit of such an evaluation and also any future perspective that present work may assist.

The authors are not planning any more work in this area. We have added a discussion to indicate the practical relevance may differ by breed LN 229-230 & 238-240.